# External Validation of Equations to Estimate Resting Energy Expenditure in 2037 Children and Adolescents with and 389 without Obesity: A Cross-Sectional Study

**DOI:** 10.3390/nu12051421

**Published:** 2020-05-14

**Authors:** Giorgio Bedogni, Simona Bertoli, Ramona De Amicis, Andrea Foppiani, Alessandra De Col, Gabriella Tringali, Nicoletta Marazzi, Valentina De Cosmi, Carlo Agostoni, Alberto Battezzati, Alessandro Sartorio

**Affiliations:** 1Clinical Epidemiology Unit, Liver Research Center, Building Q, AREA Science Park, Strada Statale 14 km 163.5, Basovizza, 34012 Trieste, Italy; 2International Center for the Assessment of Nutritional Status (ICANS), University of Milan, 20133 Milan, Italy; simona.bertoli@unimi.it (S.B.); ramona.deamicis@unimi.it (R.D.A.); andrea.foppiani@unimi.it (A.F.); alberto.battezzati@unimi.it (A.B.); 3Istituto Auxologico Italiano, IRCCS, Laboratory of Nutrition and Obesity Research, 20145 Milan, Italy; 4Istituto Auxologico Italiano, IRCCS, Experimental Laboratory for Auxo-Endocrinological Research, 20145 Milan and 28921 Verbania, Italy; a.decol@auxologico.it (A.D.C.); g.tringali@auxologico.it (G.T.); n.marazzi@auxologico.it (N.M.); sartorio@auxologico.it (A.S.); 5Pediatric Medium Intensity Care Unit, Fondazione IRCCS Cà Granda Ospedale Maggiore Policlinico, 20122 Milan, Italy; valentina.decosmi@gmail.com (V.D.C.); carlo.agostoni@unimi.it (C.A.); 6Department of Clinical Sciences and Community Health (DISCCO), University of Milan, 20122 Milan, Italy; 7Istituto Auxologico Italiano, IRCCS, Division of Auxology, 28921 Verbania, Italy

**Keywords:** obesity, children, adolescents, resting energy expenditure, indirect calorimetry, prediction equations

## Abstract

We performed an external cross-validation study of 10 equations to estimate resting energy expenditure (REE) in 2037 children with and 389 without obesity. Inclusion criteria were Caucasian ethnicity, age ≤ 18 years, and availability of REE. REE was measured using indirect calorimetry. The correct classification fraction (CCF) of an equation was defined as the fraction of subjects whose estimated REE was within 10% of measured REE. The Molnár equation was the most accurate REE prediction equation with CCFs of 0.70 (95% CI 0.65 to 0.76) in girls without obesity, 0.64 (95% CI 0.61 to 0.66) in girls with obesity, 0.76 (95% CI 0.67 to 0.83) in boys without obesity, and 0.66 (95% CI 0.63 to 0.69) in boys with obesity. The Mifflin equation was the second most accurate equation with CCFs of 0.67 (95% CI 0.61 to 0.73) in girls without obesity, 0.61 (95% CI 0.58 to 0.64) in girls with obesity, 0.75 (95% CI 0.66 to 0.82) in boys without obesity, and 0.66 (95% CI 0.63 to 0.69) in boys with obesity.

## 1. Introduction

The assessment of resting energy expenditure (REE) is central to the delivery of personalized weight loss programs. REE is in fact the starting point for the calculation of total energy expenditure (TEE), which is used to estimate the energy deficit necessary to lose weight [1]. The reference method for the assessment of REE is indirect calorimetry (IC) [2]. However, IC is expensive and available only in selected centers, and thus REE is usually estimated from prediction equations developed using IC as the reference method [1].

Persons with obesity are thought to need population-specific equations because of their peculiar body composition. As compared to persons with normal weight, those with non-syndromic obesity have in fact a larger fat-free mass (FFM) and therefore a higher REE as measured by IC (mREE). Nonetheless, when standardized on body weight, REE is generally lower in persons with obesity because their body mass is made mostly of fat mass (FM) [1].

In an external cross-validation study of REE prediction equations performed in 14,952 adults with obesity [3], we have recently shown that, independently of the chosen equation, the accuracy of estimated REE (eREE) decreases for increasing levels of body mass index (BMI). In our study of adults with obesity [3], equations not specifically developed for persons with obesity, such as the Mifflin equation [4] and the Henry equation based on weight (Wt) [5], gave the most accurate estimates of REE.

There are few external cross-validation studies of REE prediction equations in children or adolescents with obesity, most notably those of Steinberg et al. [6] and Marra et al. [7]. Steinberg et al. studied a sample of 226 U.S. adolescents with a mean age of 16 years [6] and found that the Mifflin equation [4] had the highest correct classification fraction (CCF), that is, the highest fraction of subjects whose eREE was within 10% of mREE (0.61, 95% CI 0.54 to 0.67) [6]. The second most accurate equation in their study was the Molnár equation [8], which had a CCF of 0.54 (95% CI 0.47 to 0.60) [6]. It is very interesting that the Mifflin equation [4], developed for a mixed population of adults with and without obesity, gave the most accurate estimate of REE in the children with obesity studied by Steinberg et al. [6]. Marra et al. [7] studied a sample of 109 Italian adolescents with a mean age of 16 years and found that the Lazzer equation [9] had the highest CCF (0.65, 95% CI 0.55 to 0.74), whereas the same equation [9] performed much less well in the study of Steinberg et al. [6], with a CCF of 0.36 (95% CI 0.29 to 0.42).

The aim of the present study was to externally cross-validate 10 commonly used REE prediction equations based on sex, age, weight, and height [4,5,8,10,11,12,13,14] in children with obesity using children without obesity as the comparator group. We choose these equations because (1) they are based on predictors readily available in clinical practice (age, sex, weight, and height) [1], (2) they are suggested by pediatric scientific societies [15,16], and (3) they were cross-validated by Steinberg et al. [6] and/or Marra et al. [7]. We did not include the Lazzer equation [9] because it was developed at the Istituto Auxologico Italiano, and thus its validation cannot be considered external [3]. We were especially interested to test whether the Mifflin equation [4], which has the highest CCF in adults with obesity [1,3], performed as well in our children and adolescents with obesity as it did in the adolescents studied by Steinberg et al. [6].

## 2. Materials and Methods

### 2.1. Study Design

We retrospectively collected the data of consecutive children and adolescents followed between January 2009 and June 2017 at the International Center for the Assessment of Nutritional Status (ICANS, Milan, Italy; an academic research center in agreement with the Italian National Health System) and at the Division of Auxology, Istituto Auxologico Italiano (Verbania, Italy; a private scientific institute for research and care in agreement with the Italian National Health System). The children with overweight and obesity measured at ICANS and at the Istituto Auxologico Italiano were measured at the inception of a weight loss program, whereas the children without obesity were measured at ICANS at the inception of a nutrition counseling program. We also retrospectively collected the data of a sample of healthy children studied at the Pediatric Medium Intensity Care Unit of Cà Granda (Milan, Italy; a public scientific institute for research and care in agreement with the Italian National Health System) during the year 2016. The inclusion criteria for the study were (1) Caucasian ethnic group, (2) age ≤ 18 years, and (3) availability of REE measured with a Sensor Medics Vmax 29 metabolic cart. The exclusion criteria were (1) syndromic obesity [17]; (2) dysthyroidism; (3) use of drugs known to affect energy expenditure, such as levothyroxine, and; (4) respiratory quotient (RQ) < 0.67 or > 1.3 [18]. The study was approved by the Ethical Committee of the Istituto Auxologico Italiano (research project code 01C621, acronym BEEOBCHILD) and was conducted in accordance with the 1975 Declaration of Helsinki, as revised in 2013. The parents or the legal guardians of the subjects or the subjects themselves when aged 18 years gave the written informed consent to participate in the study.

### 2.2. Anthropometry

Wt and height (Ht) were measured following international guidelines [19]. Body mass index (BMI) was calculated as weight (kg)/height (m)^2^. Standard deviations scores (SDS), that is, z-scores, of weight, height, and BMI for sex and age were calculated using Italian growth data [20]. Underweight was defined as BMI SDS < −1.644, normal weight as −1.644 ≤ BMI SDS < 1.036, overweight as 1.036 ≤ BMI SDS < 1.644, and obesity as BMI SDS ≥ 1.644.

### 2.3. REE Measurement

In each center, REE was measured between 8:00 and 10:00 a.m. in thermoneutral conditions using a Sensor Medics Vmax 29 (Yorba Linda, CA, US) metabolic cart equipped with a canopy. The gas analyzers of the metabolic cart were calibrated before each test using a reference gas mixture made of 15% O_2_ and 5% CO_2_. The subjects were in the fasting state for at least 8 h, that is, from at least 0:00 a.m.; had refrained from physical activity for at least 24 h; and had been waiting for at least 30 min in the sitting position before REE measurement. REE was measured in the supine position for at least 30 min, including an acclimation period of 10 min. The data relative to the acclimation period were discarded. The steady state was defined as at least 5 min with less than 5% variation in RQ, less than 10% variation in O_2_ consumption, and less than 10% variation in minute ventilation [18]. After the steady state was reached, O_2_ consumption and CO_2_ production were recorded at intervals of 1 min for at least 20 min and averaged over the whole period. REE was calculated from O_2_ consumption and CO_2_ production using the Weir equation [21]. Each metabolic cart underwent an ethanol burning test at least one time per year during the study period [3].

### 2.4. REE Estimation

REE was estimated using the following equations, given in Appendix A: (1) Harris–Benedict equation [10], (2) World Health Organization (WHO) equation based on Wt [13], (3) Schofield equation based on Wt [12], (4) Schofield equation based on Wt and Ht [12], (5) Henry equation [5], (6) Institute of Medicine (IOM) equation for children with normal weight (NW) [11], (7) IOM equation for children with overweight and obesity (OW and OB) [11], (8) Molnár equation [8], (9) Müller equation [14], and (10) Mifflin equation [4]. Equations (1)–(6) are suggested by pediatric scientific societies [15,16], Equations (7)–(9) included children and adolescents with obesity, and Equation (10) was developed for adults but was shown to perform well in adolescents with obesity [6]. All the equations were applied irrespective of the age for which they were developed. This implies that the Molnár equation [8] was applied also to subjects aged < 10 or > 16 years, and the Mifflin equation [4] to subjects aged < 18 years. All the other equations were appropriate for the age of the study subjects.

### 2.5. Statistical Analysis

Most continuous variables were not Gaussian-distributed and all are reported as median (50th percentile) and interquartile range (IQR; 25th and 75th percentiles). Categorical variables are reported as the number and proportion of subjects with the characteristic of interest. Within-group (girls without obesity, girls with obesity, boys without obesity, and boys with obesity) between-method (eREE vs. mREE) comparisons were performed using median regression with heteroskedasticity-robust standard errors [22,23]. Bland–Altman plots of the absolute bias (eREE−mREE) vs. the average bias [(eREE + mREE)/2] and of the percent bias [(eREE−mREE)/mREE] × 100 vs. the average bias were used to investigate the presence of proportional bias [3,24]. The mean absolute percent error (MAPE) was calculated as (|(mREE−eREE)/mREE|) × 100. The correct classification fraction (CCF), which is the main outcome of the present study, was defined as the fraction of subjects whose eREE was ≤ 10% of mREE [1]. The CCF varies from 0 (0%) to 1 (100%) and its 95% confidence intervals were computed using the “exact” Clopper–Pearson approximation [25]. Statistical analysis was performed using Stata 16.1 (Stata Corporation, College Station, TX, USA).

## 3. Results

### 3.1. Study Population

Table 1 gives the features of the 2426 children stratified by sex and obesity. The children had a median (IQR) age of 15 (13; 16) years, and 60.5% of them were girls. Children with underweight made up 0.2% of the study population; children with normal weight, 9.9%; children with overweight, 5.9%; and children with obesity, 84.0%. As expected, children with obesity were enrolled mostly at the Division of Auxology of the Istituto Auxologico Italiano, and children without obesity at ICANS and Cà Granda. Median eREE was significantly different from median mREE for all strata of sex and obesity in all cases (*p* < 0.001, median regression).

### 3.2. Accuracy of the REE Prediction Equations

Table 2 gives the absolute bias (median and IQR), the percent bias (median and IQR), the MAPE (mean), and the CCF (number and fraction) of the REE prediction equations in the 2426 study children stratified by sex and obesity. The findings obtained by percent bias, MAPE, and CCF are coherent in showing that the most accurate equations are those of Molnár [8] and Mifflin [4].

Appendix B gives the Bland–Altman plots of the percent bias vs. the average bias. Not surprisingly, proportional bias was detected for most equations [3,24]. Because of this finding, the Bland–Altman limits of agreement were not calculated.

To help to visualize the results, Figure 1 plots the median percent bias of the REE prediction equations in the 2426 children stratified by sex and obesity. Using this criterion, the best equation is that with the dot nearest to the 0 value of the *y*-axis [3].

Figure 2 plots the CCF of the REE prediction equations in the 2426 children stratified by sex and obesity. Using this criterion, which is the main outcome of the present study because of its clinical relevance [1], the best equation is that with the highest CCF [3].

In the girls without obesity, the Molnár equation had the highest CCF (0.70, 95% CI 0.65 to 0.76), followed by the IOM NW equation (0.67, 95% CI 0.62 to 0.73) and the Mifflin equation (0.67, 95% CI 0.61 to 0.73). In the girls with obesity, the Molnár equation had the highest CCF (0.64, 95% CI 0.61 to 0.66), followed by the Harris–Benedict equation (0.62, 95% CI 0.59 to 0.65), the IOM NW equation (0.62, 95% CI 0.59 to 0.65), the Schofield Wt and Ht equation (0.61, 95% CI 0.58 to 0.64), and the Mifflin equation (0.61, 95% CI 0.58 to 0.64). In the boys without obesity, the Molnár equation had the highest CCF (0.76, 95% CI 0.67 to 0.83), strictly followed by the Mifflin equation (0.75, 95% CI 0.66 to 0.82). In the boys with obesity, the Molnár equation had the highest CCF (0.66, 95% CI 0.63 to 0.69), again strictly followed by the Mifflin equation (0.64, 95% CI 0.60 to 0.67).

## 4. Discussion

### 4.1. Main Finding

In the largest study performed thus far in children and adolescents with obesity, we cross-validated 10 REE prediction equations based on sex, age, weight, and height [4,5,8,10,11,12,13,14] using IC as the reference method. The Molnár equation [8] was the most accurate REE prediction equation, independent of sex and obesity.

### 4.2. Strengths and Limitations

A clear strength of the present study is the availability of a high number of children with obesity (*n* = 2037). Our sample was in fact 9 times larger than that of Steinberg et al. [6], and 18 times larger than that of Marra et al. [7]. Another strength of the present study, as compared to other studies [6,7], is the availability of a control group of children without obesity (*n* = 389) [3].

The present study has nonetheless some limitations. The first limitation is that we did not cross-validate REE equations using FFM as predictor. There could have been some gain in predicting REE from FFM, and possibly FM, provided that some methodological precautions are taken [26]. However, FFM is often estimated, most commonly by bioelectrical impedance analysis (BIA), and not measured, and the usefulness of such a “doubly indirect” estimation of REE is doubtful [1]. Using purposely developed BIA equations to estimate FFM, we found that anthropometry- and FFM-based equations were similarly accurate at estimating REE in children with severe obesity [9], confirming the findings of Müller et al. [15] in children and adolescents with and without obesity. Our findings are also in agreement with those of Marra et al. [27], who found that raw BIA predictors (whole body impedance and phase angle) were not superior to anthropometric indicators at estimating REE. The second limitation is that we studied Caucasian children only, the reason being that non-Caucasian individuals with obesity account for less than 2% of the persons currently followed at the Istituto Auxologico Italiano, that is, the study center that enrolled most study subjects [3]. The third limitation is that our children with obesity were studied in tertiary care centers, and thus our findings may not extend to the general population. If one considers, however, that the median SDS of BMI in our children with obesity was 2.9, corresponding to the 99.8th percentile of the reference distribution [20], it should be obvious that there is no possibility of performing a cross-validation study like the present one outside tertiary care centers [3,6,7].

### 4.3. Molnár REE Prediction Equation

The Molnár equation [8] had the highest CCF in all strata of sex and obesity (Table 2 and Figure 2). However, a comparison of the CCFs of children with and without obesity must consider the different number of subjects available in each stratum. In this respect, the lower 95% CI of the CCF of the Molnár equation, which takes into account sample size and can be regarded as the worst-case scenario for practice, was similar in girls and boys without obesity (0.65 vs. 0.67) and in girls and boys with obesity (0.61 vs. 0.63). Thus, the Molnár equation is more accurate in children without than in those with obesity. Even if the Molnár equation had the second highest CCF in the study of children with obesity performed by Steinberg et al. [6], it was much lower (0.54, 95% CI 0.47 to 0.60) than that obtained in our girls and boys with obesity. The CCF of the Molnár equation was even lower in the study of Marra et al. [7] (0.45, 95% CI 0.35 to 0.58). Because the Molnár equation was developed in children aged 10 to 16 years, and in the present study we enrolled children aged 5 to 18 years, we tested whether the CCF of the Molnár equation remained the same in the subsample of children aged 10 to 16 years, and we found it to be so (data not shown). Thus, the better performance of the Molnár equation in the present study as compared to previous studies [6,7] cannot be attributed to the different age distribution from the original study.

### 4.4. Mifflin REE Prediction Equation

The Mifflin equation [4] was shown by a recent metanalysis to have the highest CCF in adults with obesity [1], and a recent cross-validation study performed by our group in 14,952 adults with obesity confirmed this finding [3]. In the study of Steinberg et al. [6], the Mifflin equation had the highest CCF (0.61, 95% CI 0.54 to 0.67) in adolescents with obesity. In the present study, we obtained CCFs of 0.61 (95% CI 0.58 to 0.64) and 0.64 (95% CI 0.60 to 0.67) in girls and boys with obesity, respectively. The Mifflin equation performed well also in girls without obesity (CCF = 0.67, 95% CI 0.61 to 0.73), where it was virtually equivalent to the IOM NW equation (Table 2 and Figure 2). Moreover, the Mifflin equation performed only slightly less well than the Harris–Benedict, IOM NW, and Schofield Wt and Ht equations in our girls with obesity (CCF = 0.61, 95% CI 0.58 to 0.64). As done above for the Molnár equation, if we look at the lower 95% CI of the Mifflin equation, which takes into account sample size and can be regarded as the worst-case scenario for practice, it was similar in girls without obesity (0.61), girls with obesity (0.58), and boys with obesity (0.60), but it was higher (0.66) in boys without obesity. Thus, the Mifflin equation appears to be more accurate in male boys without obesity. It should be added that the CCF of the Mifflin equation obtained in our children with obesity is similar to that which we obtained in adults with class II obesity (0.63%, 95% CI 0.61 to 0.65) [3]. We thus confirm the suggestion made by Steinberg et al. [6] that, even though the Mifflin equation was developed in a mixed sample of adults with and without obesity, it performs better than other apparently more population-specific equations in children with obesity. Considering that the Mifflin equation works well in both adults and children with and without obesity [1,3,6], it has a clear potential for being employed as REE prediction equation, independent of age and obesity.

### 4.5. Other REE Prediction Equations

As compared to previous cross-validation studies of REE prediction equations in children and adolescents with obesity [6,7], we obtained more optimistic estimates of the CCF for many equations. However, we confirm that most equations have a low CCF and are not suitable for practical use (Figure 1). We wish nonetheless to point out that the Harris–Benedict equation [10], developed slightly more than 100 years ago on adults and often considered “outdated”, was better or at least no worse than other equations that are considered more population-specific, such as the IOM OW and OB equation in girls and boys with obesity (Table 2 and Figure 2). Moreover, in girls with obesity, the Harris–Benedict equation was nearly as accurate as the Molnár equation. We believe that the high body weight of our children is the most likely explanation for this finding because it is our experience that the Harris–Benedict equation performs much less well in lean children [24]. Although we would not suggest replacing the Molnár or Mifflin equations with the Harris–Benedict equation, we believe that this finding is noteworthy.

## 5. Conclusions

The Molnár equation was found to be the most accurate equation for the estimation of REE in children with and without obesity, and the Mifflin equation provided another good choice. It must be noted, however, that the use of a prediction equation involves the loss of individual information because of data reduction. Thus, any equation will be, in the end, unsatisfactory for some individuals. Having an idea of the number of such individuals is possibly the most important practical criterion to guide the choice of an REE prediction equation for use at the individual level. In this respect, further cross-validation studies and metanalyses of available studies are needed to understand what are the best REE equations for children with obesity [1].

## Figures and Tables

**Figure 1 nutrients-12-01421-f001:**
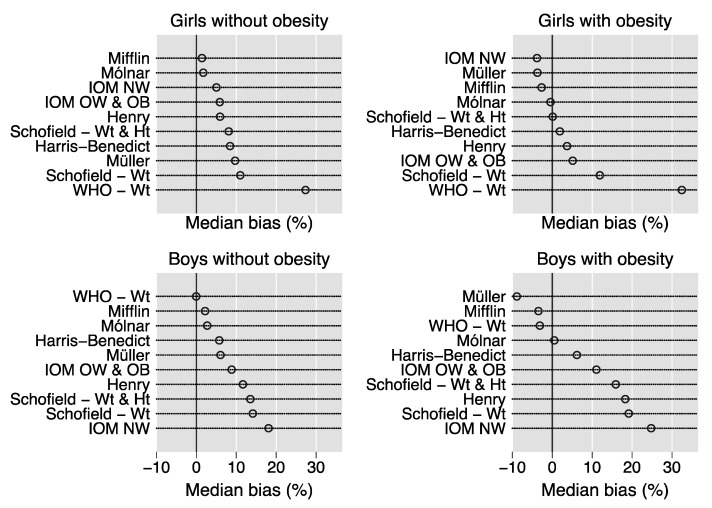
Median percent bias of the REE prediction equations. Dot charts showing the median percent bias of the REE prediction equations in the 2426 children stratified by sex and obesity. Percent bias is calculated as [(estimated REE-measured REE)/measured REE]*100. Abbreviations: WHO = World Health Organization; Wt = weight; Ht = height; IOM = Institute of Medicine; NW = (equation for) children with normal weight; OW = (equation for) children with overweight; OB = (equation for) children with obesity.

**Figure 2 nutrients-12-01421-f002:**
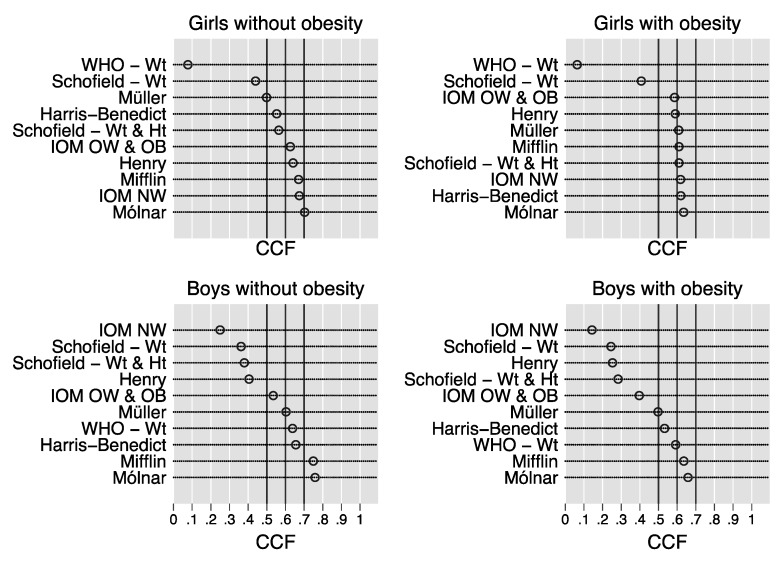
Correct classification fraction of the REE prediction equations. Dot charts showing the correct classification fraction of the REE prediction equations in the 2426 children stratified by sex and obesity. The correct classification fraction of an equation is defined as the fraction of subjects whose estimated resting energy expenditure is within 10% of measured resting energy expenditure and varies from 0 (0%) to 1 (100%). Abbreviations: WHO = World Health Organization; Wt = weight; Ht = height; IOM = Institute of Medicine; NW = (equation for) children with normal weight; OW = (equation for) children with overweight; OB = (equation for) children with obesity.

**Table 1 nutrients-12-01421-t001:** Sex, age, anthropometry, and resting energy expenditure of the study population.

	Girls with Obesity	Boys without Obesity	Boys with Obesity	All Children	Girls with Obesity
	*N* = 273	*N* = 1194	*N* = 116	*N* = 843	*N* = 2426
Center					
Istituto Auxologico Italiano-Auxology	4 (1.5%)	1070 (89.6%)	2 (1.7%)	761 (90.3%)	1837 (75.7%)
ICANS	230 (84.2%)	123 (10.3%)	80 (69.0%)	81 (9.6%)	514 (21.2%)
Cà Granda	39 (14.3%)	1 (0.1%)	34 (29.3%)	1 (0.1%)	75 (3.1%)
Sex					
Female	273 (100.0%)	1194 (100.0%)	0 (0.0%)	0 (0.0%)	1467 (60.5%)
Male	0 (0.0%)	0 (0.0%)	116 (100.0%)	843 (100.0%)	959 (39.5%)
Age (years)	15 (12; 16)	15 (13; 16)	12 (10; 15)	15 (12; 16)	15 (13; 16)
Age category					
Children (< 13 years)	103 (37.7%)	295 (24.7%)	76 (65.5%)	264 (31.3%)	738 (30.4%)
Adolescents (> 13 years)	170 (62.3%)	899 (75.3%)	40 (34.5%)	579 (68.7%)	1688 (69.6%)
Weight (kg)	59 (47; 66)	91 (80; 103)	56 (37; 67)	100 (84; 118)	89 (72; 106)
Weight (SDS Cacciari 2006)	0.80 (0.05; 1.29)	2.97 (2.38; 3.53)	0.78 (0.04; 1.28)	2.81 (2.30; 3.36)	2.71 (1.97; 3.33)
Height (m)	1.61 (1.52; 1.66)	1.60 (1.55; 1.65)	1.54 (1.41; 1.72)	1.68 (1.58; 1.75)	1.62 (1.55; 1.69)
Stature (SDS Cacciari 2006)	0.56 (−0.17; 1.21)	0.30 (−0.38; 1.04)	0.81 (−0.10; 1.47)	0.41 (−0.24; 1.14)	0.39 (−0.28; 1.14)
BMI (kg/m^2^)	23 (20; 25)	35 (31; 39)	21 (18; 24)	35 (31; 40)	34 (29; 39)
BMI (SDS Cacciari 2006)	0.80 (−0.14; 1.19)	2.80 (2.37; 3.23)	0.69 (−0.30; 1.26)	2.84 (2.35; 3.32)	2.65 (2.08; 3.18)
BMI class (Cacciari 2006)					
Underweight	4 (1.5%)	0 (0.0%)	2 (1.7%)	0 (0.0%)	6 (0.2%)
Normal weight	168 (61.5%)	0 (0.0%)	71 (61.2%)	0 (0.0%)	239 (9.9%)
Overweight	101 (37.0%)	0 (0.0%)	43 (37.1%)	0 (0.0%)	144 (5.9%)
Obesity	0 (0.0%)	1194 (100.0%)	0 (0.0%)	843 (100.0%)	2037 (84.0%)
Respiratory quotient	0.83 (0.79; 0.87)	0.83 (0.77; 0.88)	0.85 (0.82; 0.90)	0.82 (0.77; 0.88)	0.83 (0.78; 0.88)
mREE (kcal/day)	1336 (1184; 1448)	1719 (1552; 1925)	1402 (1208; 1641)	2085 (1812; 2356)	1756 (1510; 2055)
mREE (kcal/kg weight/day)	23 (21; 26)	19 (17; 21)	26 (24; 33)	21 (19; 23)	20 (18; 23)
eREE WHO-Wt (kcal/day)	1712 * (1493; 1834)	2273 * (2073; 2491)	1441 * (1213; 1564)	1975 * (1784; 2187)	2079 * (1815; 2336)
eREE Schofield-Wt (kcal/day)	1489 * (1327; 1583)	1915 * (1769; 2078)	1648 * (1311; 1839)	2432 * (2144; 2740)	1968 * (1729; 2319)
eREE Schofield-Wt and Ht (kcal/day)	1448 * (1321; 1525)	1716 * (1612; 1830)	1644 * (1312; 1841)	2370 * (2099; 2667)	1779 * (1604; 2168)
eREE Henry (kcal/day)	1418 * (1287; 1497)	1776 * (1653; 1910)	1608 * (1267; 1806)	2423 * (2121; 2742)	1842 * (1633; 2221)
eREE Harris–Benedict (kcal/day)	1455 * (1329; 1525)	1754 * (1642; 1870)	1540 * (1227; 1751)	2190 * (1938; 2448)	1791 * (1605; 2067)
eREE Molnár (kcal/day)	1320 * (1182; 1435)	1706 * (1565; 1862)	1491 * (1204; 1687)	2070 * (1835; 2305)	1750 * (1529; 2010)
eREE IOM NW (kcal/day)	1411 * (1283; 1484)	1651 * (1553; 1761)	1693 * (1349; 1916)	2556 * (2242; 2898)	1735 * (1551; 2300)
eREE IOM OW and OB (kcal/day)	1423 * (1259; 1510)	1804 * (1669; 1954)	1543 * (1266; 1720)	2279 * (2021; 2572)	1857 * (1632; 2173)
eREE Mifflin (kcal/day)	1367 * (1211; 1459)	1682 * (1554; 1817)	1475 * (1228; 1690)	1990 * (1778; 2194)	1714 * (1520; 1944)
eREE Müller (kcal/day)	1471 * (1341; 1558)	1660 * (1566; 1761)	1514 * (1311; 1711)	1875 * (1714; 2033)	1687 * (1552; 1847)

* *p* < 0.001 vs. mREE within all strata of sex and obesity (median regression). Continuous variables are reported as 50th (median) and 25th and 75th percentiles (interquartile range, within brackets). Discrete variables are reported as the number and proportion (within brackets) of subjects with the characteristic of interest. Abbreviations: *N* = number of children; SDS = standard deviation scores (z-scores); BMI = body mass index; mREE = measured resting energy expenditure (REE); eREE = estimated resting energy expenditure; Wt = weight; Ht = height; WHO = World Health Organization; IOM = Institute of Medicine; NW = (equation for) children with normal weight; OW = (equation for) children with overweight; OB = (equation for) children with obesity.

**Table 2 nutrients-12-01421-t002:** Absolute bias, percent bias, mean absolute percent error, and correct classification fraction of the REE prediction equations.

	Girls without Obesity	Girls with Obesity	Boys without Obesity	Boys with Obesity	All Children
	*N* = 273	*N* = 1194	*N* = 116	*N* = 843	*N* = 2426
WHO-Wt bias (kcal/day)	364 (261; 447)	548 (385; 715)	−1 (−94; 80)	−64 (−236; 88)	318 (−1; 557)
WHO-Wt bias (%)	27 (20; 35)	32 (22; 43)	0 (−6; 7)	−3 (−11; 5)	20 (0; 35)
WHO-Wt MAPE (%, mean)	29	33	10	10	24
WHO-Wt CCF	21 (7.7%)	77 (6.4%)	74 (63.8%)	499 (59.2%)	671 (27.7%)
Schofield-Wt bias (kcal/day)	148 (64; 224)	202 (66; 334)	197 (66; 287)	373 (200; 530)	236 (95; 391)
Schofield-Wt bias (%)	11 (5; 18)	12 (4; 21)	14 (5; 20)	19 (10; 27)	14 (6; 23)
Schofield-Wt MAPE (%, mean)	15	15	16	20	17
Schofield-Wt CCF	120 (44.0%)	486 (40.7%)	42 (36.2%)	207 (24.6%)	855 (35.2%)
Schofield-Wt and Ht bias (kcal/day)	108 (26; 186)	2 (−139; 124)	186 (65; 269)	309 (148; 459)	109 (−48; 276)
Schofield-Wt and Ht bias (%)	8 (2; 15)	0 (−7; 8)	13 (5; 19)	16 (7; 24)	7 (−3; 17)
Schofield-Wt and Ht MAPE (%, mean)	13	9	15	17	13
Schofield-Wt and Ht CCF	154 (56.4%)	729 (61.1%)	44 (37.9%)	239 (28.4%)	1166 (48.1%)
Henry bias (kcal/day)	80 (1; 164)	64 (−69; 191)	164 (53; 264)	356 (178; 525)	143 (−2; 321)
Henry bias (%)	6 (0; 13)	4 (−4; 12)	12 (3; 18)	18 (9; 27)	9 (0; 19)
Henry MAPE (%, mean)	12	10	14	20	14
Henry CCF	175 (64.1%)	705 (59.0%)	47 (40.5%)	214 (25.4%)	1141 (47.0%)
Harris–Benedict bias (kcal/day)	113 (36; 194)	34 (−96; 153)	81 (−24; 166)	124 (−43; 263)	77 (−58; 199)
Harris–Benedict bias (%)	8 (2; 16)	2 (−5; 10)	6 (−2; 11)	6 (−2; 14)	4 (−3; 12)
Harris–Benedict MAPE (%, mean)	14	9	10	11	10
Harris–Benedict CCF	151 (55.3%)	740 (62.0%)	76 (65.5%)	449 (53.3%)	1416 (58.4%)
Molnár bias (kcal/day)	22 (−67; 87)	−7 (−138; 124)	36 (−52; 113)	9 (−148; 146)	5 (−125; 126)
Molnár bias (%)	2 (−5; 8)	0 (−8; 7)	3 (−4; 8)	1 (−7; 8)	0 (−7; 8)
Molnár MAPE (%, mean)	11	9	9	9	9
Molnár CCF	192 (70.3%)	758 (63.5%)	88 (75.9%)	555 (65.8%)	1593 (65.7%)
IOM NW bias (kcal/day)	68 (−7; 150)	−64 (−201; 49)	260 (141; 358)	491 (298; 675)	87 (−86; 360)
IOM NW bias (%)	5 (−1; 12)	−4 (−11; 3)	18 (10; 26)	25 (14; 34)	6 (−5; 21)
IOM NW MAPE (%, mean)	11	9	20	25	16
IOM NW CCF	184 (67.4%)	738 (61.8%)	29 (25.0%)	121 (14.4%)	1072 (44.2%)
IOM OW and OB bias (kcal/day)	78 (18; 169)	87 (−46; 212)	127 (15; 207)	229 (61; 368)	126 (1; 262)
IOM OW and OB bias (%)	6 (1; 14)	5 (−2; 13)	9 (1; 15)	11 (3; 19)	7 (0; 16)
IOM OW and OB MAPE (%, mean)	12	10	12	14	12
IOM OW and OB CCF	171 (62.6%)	700 (58.6%)	62 (53.4%)	335 (39.7%)	1268 (52.3%)
Mifflin bias (kcal/day)	19 (−71; 97)	−46 (−180; 73)	32 (−46; 104)	−72 (−226; 58)	−39 (−175; 76)
Mifflin bias (%)	1 (−6; 8)	−3 (−10; 5)	2 (−3; 8)	−3 (−11; 3)	−2 (−9; 5)
Mifflin MAPE (%, mean)	10	9	8	9	9
Mifflin CCF	183 (67%)	728 (61%)	87 (75%)	536 (64%)	1534 (63%)
Müller bias (kcal/day)	130 (47; 204)	−65 (−197; 60)	85 (1; 163)	−180 (−364; −40)	−65 (−243; 72)
Müller bias (%)	10 (3; 18)	−4 (−11; 4)	6 (0; 13)	−9 (−16; −2)	−4 (−12; 5)
Müller MAPE (%, mean)	14	9	11	12	11
Müller CCF	136 (50%)	726 (61%)	70 (60%)	419 (50%)	1351 (56%)

Continuous variables are reported as 50th (median) and 25th and 75th percentiles (interquartile range, within brackets) except for mean absolute percent error, which is reported as mean. Discrete variables are reported as the number and proportion (within brackets) of subjects with the characteristic of interest. Bias is calculated as [(estimated REE − measured REE)/measured REE] and percent bias as [(estimated REE − measured REE)/measured REE]*100. Abbreviations: *N* = number of children; MAPE = mean absolute percent error; WHO = World Health Organization; SDS = standard deviation scores (z-scores); Wt = weight; CCF = correct classification fraction; Ht = height; IOM = Institute of Medicine; NW = (equation for) children with normal weight; OW = (equation for) children with overweight; OB = (equation for) children with obesity. The different metrics agree to identify the Molnár and Muller equations as the most accurate equations.

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
