# Peer review of "External Validation of Equations to Estimate Resting Energy Expenditure in 2037 Children and Adolescents with and 389 without Obesity: A Cross-Sectional Study"

_nutrients, 2020, doi:10.3390/nu12051421_

Round 1
Reviewer 1 Report
The major drawback of this study is that the authors have used REE prediction algorithms which are inadequate for children and adolescents (e.g. the reference population of the Harris Benedict equation is an adult population). Thus, refering to algorithms generated for adults is misleading. In addition some highly cited studies including adolescents and slso obese adolescents (e.g. Müller et al, AJCN 2004) had been ignored. As a general rule the impressive data base of the present study has limited value because of the lack of any estimate of body composition.
Author Response
*Reviewer 1*
Q1.1: The major drawback of this study is that the authors have used REE prediction algorithms which are inadequate for children and adolescents (e.g. the reference population of the Harris Benedict equation is an adult population). Thus, refering to algorithms generated for adults is misleading.
A1.1: The reason why we included the Harris-Benedict equation is that it is among the equations presently suggested by Scientific Societies for children. The Harris-Benedict equation was also included in the cross-validation study of adolescents with obesity made by Steinberg et al. [1]. These are the reasons why we thought important to cross-validate the Harris-Benedict equation in our study population. We were the first to be surprised by the fact that the Harris-Benedict equation was not worse and, in some cases, better than supposedly more population-specific equations. Taking into consideration the comment raised by the Reviewer, we have now added a comment and provided a possible explanation for it in the Discussion. Moreover, as reported in the text “We were especially interested to test whether the Mifflin equation, which has the highest correct classification fraction in adults with obesity, performed as well in our children with obesity as it did in those studied by Steinberg et al. [1]”. We believe that this evidence justifies our choice of including selected REE equations not specific for children.
Q1.2: In addition some highly cited studies including adolescents and also obese adolescents (e.g. Müller et al., AJCN 2004) had been ignored.
A1.2: We thank the Reviewer for the suggestion. We added the Müller equation to the list of cross-validated equations [2]. Please see the revised MS.
Q1.3: As a general rule the impressive data base of the present study has limited value because of the lack of any estimate of body composition.
A1.3: We agree with the Reviewer that FFM is, at least on theoretical grounds, a better predictor of REE than body weight. This is a point that we discussed in the Introduction and that now we have addressed more extensively in the Discussion. The reasons why we did not consider FFM (and FM) are the following: 1) no direct measurement of FFM was available in any center; 2) we could estimate FFM from bioelectrical impedance analysis (BIA) in the children followed at the Istituto Auxologico Italiano but: 2a) we had already shown that even with purposely developed FFM prediction formulae based on BIA, the estimate of REE obtained from FFM is not more accurate than that obtained with the standard approach [3]; 2b) we have no similar equations for the children followed at ICANS and Cà Granda and applying BIA equations available in the literature would expose us to an unknown estimation error; 3c) in many research and clinical settings, even an indirect estimate of FFM would be difficult to obtain.” However, since the lack of body composition data might represent a limit of the study despite the above considerations, we have underlined this aspect in the revised text:
“The present study has nonetheless some limitations. The first limitation is that we did not cross-validate REE equations using FFM as predictor. There could be some gain in predicting REE from FFM, and possibly FM, provided that some methodological precautions are taken [4]. However, FFM is often estimated, most commonly by bioelectrical impedance analysis (BIA), and not measured, and the usefulness of a such “doubly indirect” estimation of REE is doubtful [5]. Using purposely developed BIA equations to estimate FFM, we found that anthropometry- and FFM-based equations were similarly accurate at estimating REE in children with severe obesity [3]. These findings are similar to those of Marra et al. [6], who found that raw BIA predictors (whole body impedance and phase angle) were equivalent to anthropometric indicators at estimating REE.” In the revised version of the MS, we have added that also the FFM-based equation of Müller was not superior to a purposely developed equation based on weight, height and age at estimating REE in children and adolescents [2]. Please, see also A1.2. We hope that this line of reasoning makes it definitely clear why preferred to avoid reporting estimates rather than measurements of body composition.
References (for all reviewers)
- Steinberg A.; Manlhiot C.; Cordeiro K.; Chapman K.; Pencharz P. B.; McCrindle B. W.; Hamilton J. K. Determining the accuracy of predictive energy expenditure (PREE) equations in severely obese adolescents. Clin Nutr. 2017;36:1158-1164.
- Müller M. J.; Bosy-Westphal A.; Klaus S.; Kreymann G.; Lührmann P. M.; Neuhäuser-Berthold M.; Noack R.; Pirke K. M.; Platte P.; Selberg O. World Health Organization equations have shortcomings for predicting resting energy expenditure in persons from a modern, affluent population: generation of a new reference standard from a retrospective analysis of a German database of resting energy expenditure. The American journal of clinical nutrition. 2004;80:1379-1390.
- Lazzer S.; Agosti F.; De Col A.; Sartorio A. Development and cross-validation of prediction equations for estimating resting energy expenditure in severely obese Caucasian children and adolescents. Br J Nutr. 2006;96:973-979.
- Bosy-Westphal A.; Braun W.; Schautz B.; Müller M. J. Issues in characterizing resting energy expenditure in obesity and after weight loss. Front Physiol. 2013;4:47.
- Madden A. M.; Mulrooney H. M.; Shah S. Estimation of energy expenditure using prediction equations in overweight and obese adults: a systematic review. J Hum Nutr Diet. 2016;29:458-476.
- Marra M.; Cioffi I.; Sammarco R.; Santarpia L.; Contaldo F.; Scalfi L.; Pasanisi F. Are Raw BIA Variables Useful for Predicting Resting Energy Expenditure in Adults with Obesity. Nutrients. 2019;11
- Bedogni G.; Bertoli S.; Leone A.; De Amicis R.; Lucchetti E.; Agosti F.; Marazzi N.; Battezzati A.; Sartorio A. External validation of equations to estimate resting energy expenditure in 14952 adults with overweight and obesity and 1948 adults with normal weight from Italy. Clin Nutr. 2019;38:457-464.
- Agostoni C.; Edefonti A.; Calderini E.; Fossali E.; Colombo C.; Battezzati A.; Bertoli S.; Milani G.; Bisogno A.; Perrone M.; Bettocchi S.; De Cosmi V.; Mazzocchi A.; Bedogni G. Accuracy of Prediction Formulae for the Assessment of Resting Energy Expenditure in Hospitalized Children. J Pediatr Gastroenterol Nutr. 2016;63:708-712.
- Conroy R. M. What hypotheses do “nonparametric” two-group tests actually test? Stata Journal. 12:182-190.
- Marra M.; Montagnese C.; Sammarco R.; Amato V.; Della Valle E.; Franzese A.; Contaldo F.; Pasanisi F. Accuracy of predictive equations for estimating resting energy expenditure in obese adolescents. J Pediatr. 2015;166:1390-6.e1.
- Shetty P. Energy requirements of adults. Public Health Nutrition. 2005;8:994-1009.
Reviewer 2 Report
Comments to the Author
This paper aims to provide with an external cross-calibration of REE prediction equations in obese children and adolescents using indirect calorimetry as the reference method. To my knowledge, this is the largest cross-calibration study performed so far on children with obesity. Thus, this investigation adds significant data to the research available in children, making the study of interest for a careful selection of specific prediction equations, which holds crucial clinical relevance. The study comes from an experienced group, being well-designed and conducted, the manuscript is well organized, well written and the discussion is in accordance with the results reported and correctly fitted within the literature on predictive REE equations. Nonetheless, yet I would strongly suggest the inclusion of some additional statistical analysis and plots which could lead to an easier read, making the article more practical to be applied in the real-world clinical setting. Exclusively reporting and basing the conclusions on CCF seems somewhat scarce given the amount of data reported. Furthermore a more in-depth explanation of the REE measurements by indirect calorimetry should be added, as explain below.
My detailed comments follow:
Major comments
- Did the authors use the same metabolic cart and methods in the three centers? If this was not the case, did the authors cross-calibrated the metabolic carts? No data on accuracy and precision of the devices was included. Were the carts validated using ethanol burning tests? All this question should be clearly mentioned to show how valid and reliable are the indirect calorimetry measurements.
- All data referring to bias seem totally underscored in the article and are even not mentioned whatsoever in the discussion nor in the conclusions. The authors mentioned CCF as the main variable to be considered in the article, but I consider that this in compatible with also making a better use of other statistical variables. I suggest that Bland&Altman plots are included, at least in percentage (so it allows to account for body size between boys and girls) and also paralell limits of agreement could be included if percentage difference plots are used and algo V-shaped limits of agreement could be implemented (see PMID: 11978620, reference by the authors). A similar plot to that used to rank the equations by CCF could be added using the median percent bias. Given the sample size of the study (Central Limit Theorem), paired and unpaired parametric t-tests could be used to test differences between predicted and measured REE as well as to the corresponding biases. The mean absolute percentage error (MAPE) could be easily added, which would also provide with useful information of within the tables to interpret the predictive errors taking into account body size in girls and boys (|measured-predicted REE |*100/measured REE)).
- Discussion and conclusion should benefit and be reinforced by the extended statistical analysis mentioned above, as well as the conclusions.
- Most of the equations to predict REE in children and adolescent ignore the inclusion of the pubertal stage, but not all (such as Molnar´s who assessed it according to Tanner JM. Growth at adolescence, 2nd ed. Oxford: Blackwell, Scientific Publication, 1962. Whether the cross-validated equations took this into account or not when developing the equations should be mentioned and discussed.
Minor comments:
Introduction, page 3, line 101: overweight BMI SDS is reported twice instead of obese.
Author Response
*Reviewer 2*
Comment: This paper aims to provide with an external cross-calibration of REE prediction equations in obese children and adolescents using indirect calorimetry as the reference method. To my knowledge, this is the largest cross-calibration study performed so far on children with obesity. Thus, this investigation adds significant data to the research available in children, making the study of interest for a careful selection of specific prediction equations, which holds crucial clinical relevance. The study comes from an experienced group, being well-designed and conducted, the manuscript is well organized, well written and the discussion is in accordance with the results reported and correctly fitted within the literature on predictive REE equations.
Reply: We thank the reviewer for the comment.
Comment: Nonetheless, yet I would strongly suggest the inclusion of some additional statistical analysis and plots which could lead to an easier read, making the article more practical to be applied in the real-world clinical setting. Exclusively reporting and basing the conclusions on CCF seems somewhat scarce given the amount of data reported. Furthermore a more in-depth explanation of the REE measurements by indirect calorimetry should be added, as explain below.
Reply: please see our replies below.
Major comments
Q2.1: Did the authors use the same metabolic cart and methods in the three centers? If this was not the case, did the authors cross-calibrated the metabolic carts?
A2.1 Each center used its own Vmax 29 (Sensor Medics, Yorba Linda, CA) metabolic cart. We have rewritten the corresponding passage as follows:
“In each center, REE was measured between 8:00 and 10:00 AM in thermoneutral conditions using a Sensor Medics Vmax 29 (Yorba Linda, CA, US) metabolic cart equipped with a canopy”.
Q2.2: No data on accuracy and precision of the devices was included. Were the carts validated using ethanol burning tests? All this question should be clearly mentioned to show how valid and reliable are the indirect calorimetry measurements.
A2.1: Yes, the metabolic carts underwent periodic validation with the ethanol burning test. This is now stated as: “Each metabolic cart underwent an ethanol burning test at least one time per year during the study period [7]”.
Q2.3: All data referring to bias seem totally underscored in the article and are even not mentioned whatsoever in the discussion nor in the conclusions.
A2.3: We focused on the correct classification fraction (CCF) because it is more directly interpretable than absolute or percent bias [5]. CCF answers the question “Out of 100 persons, how many will have their estimated REE with 10% of measured REE?”. Of course, one cannot know whether a given individual will be correctly classified, but the CCF gives the clinician the frequency of the acceptable individual error associated with a given equation. We added, however, the requested data. Please, see below.
Q2.4 The authors mentioned CCF as the main variable to be considered in the article, but I consider that this in compatible with also making a better use of other statistical variables.
A2.4: please, see A2.3.
Q2.5 I suggest that Bland&Altman plots are included, at least in percentage (so it allows to account for body size between boys and girls) and also paralell limits of agreement could be included if percentage difference plots are used and algo V-shaped limits of agreement could be implemented (see PMID: 11978620, reference by the authors).
A2.5 As discussed in the Statistical analysis paragraph, we have drawn Bland-Altman plots since the first version of the MS. We do this routinely, for both absolute and percent bias. The problem is that the bias is non-proportional in virtually all cases, making the calculation of the limits of agreement invalid. This was largely expected [7, 8]. No commonly tried transformation, e.g. the logarithmic one, made the bias proportional. The Bland-Altman plots for the percent bias vs. the average bias are now given in Appendix 2. We used a running plot smoother to show the lack of proportional bias in virtually all cases (10 equations x 4 strata of sex and obesity).
Q2.6 A similar plot to that used to rank the equations by CCF could be added using the median percent bias.
A2.6 We thank the Reviewer for the relevant suggestion. We have added the plot of median percent bias Please, see Figure 1 of R2-MS.
Q2.7 Given the sample size of the study (Central Limit Theorem), paired and unpaired parametric t-tests could be used to test differences between predicted and measured REE as well as to the corresponding biases.
A2.7 Our aim here, as in our previous study [7] was to estimate the CCF and its precision (via its 95%CI). No hypothesis testing is needed for that and this explains why we did not report any p-value in our previous study [7]. However, following the request of the Reviewer, we have added the p-values obtained from testing the null “Is median eREE – median mREE = 0”? We have used median regression to compare the medians. We have avoided using non-parametric methods because they are usually not appropriate to compare medians [9].
Q2.8 The mean absolute percentage error (MAPE) could be easily added, which would also provide with useful information of within the tables to interpret the predictive errors taking into account body size in girls and boys (|measured-predicted REE |*100/measured REE)).
A2.8 As requested by the Reviewer, the MAPE has been added.
Q2.9 Discussion and conclusion should benefit and be reinforced by the extended statistical analysis mentioned above, as well as the conclusions.
A2.9 We have implemented the results with the above metrics but we still base our conclusion on the CCF and its 95%CI. There is of course a general agreement between the different metrics but the most useful applicable metric actually remains the CCF. It is also the one which allows the most useful comparison with the available studies [1, 5, 10].
Q2.10 Most of the equations to predict REE in children and adolescent ignore the inclusion of the pubertal stage, but not all (such as Molnar´s who assessed it according to Tanner JM. Growth at adolescence, 2nd ed. Oxford: Blackwell, Scientific Publication, 1962. Whether the cross-validated equations took this into account or not when developing the equations should be mentioned and discussed.
A2.10 We thank the Reviewer for the comment. We did not consider puberty for three reasons: 1) none of the algorithms that we and others [1, 10] choose to cross-validate included puberty. (This was a cross-validation study and not a study aimed at generating new algorithms, possibly including pubertal status); 2) pubertal status (Tanner staging) was available only for the children studied at the Istituto Auxologico Italiano (Italian Institute of Auxology), i.e. children with obesity; 3) a properly done assessment of pubertal status requires a physical examination performed by a pediatric endocrinologist, who is not available in most research and clinical settings. This implies that a formula including pubertal status, assuming of course that pubertal status adds to age predicting REE (and setting aside the likely problem of collinearity), could be used in very few clinical settings.
Minor comments:
Q2.11 Introduction, page 3, line 101: overweight BMI SDS is reported twice instead of obese.
A2.11 We thank the reviewer much for spotting the error, which was corrected.
References (for all reviewers)
- Steinberg A.; Manlhiot C.; Cordeiro K.; Chapman K.; Pencharz P. B.; McCrindle B. W.; Hamilton J. K. Determining the accuracy of predictive energy expenditure (PREE) equations in severely obese adolescents. Clin Nutr. 2017;36:1158-1164.
- Müller M. J.; Bosy-Westphal A.; Klaus S.; Kreymann G.; Lührmann P. M.; Neuhäuser-Berthold M.; Noack R.; Pirke K. M.; Platte P.; Selberg O. World Health Organization equations have shortcomings for predicting resting energy expenditure in persons from a modern, affluent population: generation of a new reference standard from a retrospective analysis of a German database of resting energy expenditure. The American journal of clinical nutrition. 2004;80:1379-1390.
- Lazzer S.; Agosti F.; De Col A.; Sartorio A. Development and cross-validation of prediction equations for estimating resting energy expenditure in severely obese Caucasian children and adolescents. Br J Nutr. 2006;96:973-979.
- Bosy-Westphal A.; Braun W.; Schautz B.; Müller M. J. Issues in characterizing resting energy expenditure in obesity and after weight loss. Front Physiol. 2013;4:47.
- Madden A. M.; Mulrooney H. M.; Shah S. Estimation of energy expenditure using prediction equations in overweight and obese adults: a systematic review. J Hum Nutr Diet. 2016;29:458-476.
- Marra M.; Cioffi I.; Sammarco R.; Santarpia L.; Contaldo F.; Scalfi L.; Pasanisi F. Are Raw BIA Variables Useful for Predicting Resting Energy Expenditure in Adults with Obesity. Nutrients. 2019;11
- Bedogni G.; Bertoli S.; Leone A.; De Amicis R.; Lucchetti E.; Agosti F.; Marazzi N.; Battezzati A.; Sartorio A. External validation of equations to estimate resting energy expenditure in 14952 adults with overweight and obesity and 1948 adults with normal weight from Italy. Clin Nutr. 2019;38:457-464.
- Agostoni C.; Edefonti A.; Calderini E.; Fossali E.; Colombo C.; Battezzati A.; Bertoli S.; Milani G.; Bisogno A.; Perrone M.; Bettocchi S.; De Cosmi V.; Mazzocchi A.; Bedogni G. Accuracy of Prediction Formulae for the Assessment of Resting Energy Expenditure in Hospitalized Children. J Pediatr Gastroenterol Nutr. 2016;63:708-712.
- Conroy R. M. What hypotheses do “nonparametric” two-group tests actually test? Stata Journal. 12:182-190.
- Marra M.; Montagnese C.; Sammarco R.; Amato V.; Della Valle E.; Franzese A.; Contaldo F.; Pasanisi F. Accuracy of predictive equations for estimating resting energy expenditure in obese adolescents. J Pediatr. 2015;166:1390-6.e1.
- Shetty P. Energy requirements of adults. Public Health Nutrition. 2005;8:994-1009.
Reviewer 3 Report
Abstract: Include selection criteria for participants.
Introduction:
- 1st paragraph: The reference 2 is very old (2005 - 15 years ago), I suggest you cite a more recent reference that states that REE is still important for energy calculation, since some authors suggest energy restriction from food intake data
- 2st paragraph: All citations are from reference 3, so I suggest that this reference is only cited at the end of the paragraph
- 3st paragraph: Include citation 4 right after "In our study of adults with obesity ... "to make it clear which study of the research group the authors are referring to.
- 4st paragraph: The equation by Steinberg et al (2017) used a sample of adolescents and not children, as written in line 63
- Improve the justification for this paper. Why invest in research on energy expenditure prediction and predictive equations for children, especially with obesity, if some guidelines do not recommend energy restriction but change in lifestyle (physical exercise and quality of diet)?
Materials and Methods: Briefly describe the locations for acquisition of participants (public? Private?)
Results:
- Table 1: I suggest presenting the classification data by age group (children and adolescents) and not for each age found
- Explain the differences between the research centers included in the study to clarify the difference in the results found
- Table 1 also contains data for mREE and eREE that were not described in the text. It is necessary to describe the main descriptive findings of mREE and eREE.
- It is necessary to improve the title of table 1. What are the characteristics of the sample studied that appear in that table? To specify.
- Text about the Table 2: Better describe the interpretation of the data. What are the main results?
Author Response
Reviewer 3
We thank the reviewer for the comments.
Q3.1 Abstract: Include selection criteria for participants.
A3.1 We thank the Reviewer for the excellent suggestion. We have added the following passage to the abstract: “Inclusion criteria were: Caucasian ethnicity, age < 18 years and availability of REE”.
Introduction:
Q3.2 - 1st paragraph: The reference 2 is very old (2005 - 15 years ago), I suggest you cite a more recent reference that states that REE is still important for energy calculation, since some authors suggest energy restriction from food intake data.
A3.2 We thank the Reviewer for the suggestion. We have now quoted Madden et al. [5] instead of Shetty et al. [11] because they explain the rationale behind the measurement of REE (and TEE) with great clarity.
Q3.3 - 2st paragraph: All citations are from reference 3, so I suggest that this reference is only cited at the end of the paragraph.
A3.3 We thank the Reviewer for the suggestion. Madden et al. [5] has been quoted only at the end of the paragraph.
Q3.4 - 3st paragraph: Include citation 4 right after "In our study of adults with obesity ... " to make it clear which study of the research group the authors are referring to.
A3.4 We thank the Reviewer for the suggestion. Bedogni et al. [7] has been moved rightly after “In our study of adults with obesity”.
Q3.5 - 4st paragraph: The equation by Steinberg et al (2017) used a sample of adolescents and not children, as written in line 63.
A3.4 We thank the Reviewer for pointing out the inconsistency, which has been now resolved [1].
Q3.6 - Improve the justification for this paper. Why invest in research on energy expenditure prediction and predictive equations for children, especially with obesity, if some guidelines do not recommend energy restriction but change in lifestyle (physical exercise and quality of diet)?
A3.6 We certainly do not oppose the idea of recommending lifestyle changes to persons with obesity. This is exactly what we do at our Institutions and the reason why our teams comprehend dietitians, physical exercise experts, and psychologists. We strongly believe, however, that personalizing such changes requires at least a rough idea of total energy expenditure as pointed out, among others, by Madden et al [5]. We are lucky of course to have REE measurements available at our Centers but the equations produced by us and others may benefit the whole community provided that one knows the limitations of any predictive equation. We prefer not to write this in the MS for two reasons: 1) this is an Open Review, which will be accessible to everyone; 2) we are not aware of any randomized trial comparing the “fully qualitative” approach vs. the more “traditional” one.
Q3.7 Materials and Methods: Briefly describe the locations for acquisition of participants (public? Private?).
A3.7 We thank the Reviewer for the suggestion. We have now specified that:
1) ICANS (Milan, Italy) is an Academic Research Center in agreement with the Italian National Health System;
2) Istituto Auxologico Italiano (Verbania, Italy) is a private Scientific Institute for Research and Care in agreement with the Italian National Health System);
3) Cà Granda (Milan, Italy) is a public Scientific Institute for Research and Care in agreement with the Italian National Health System).
Results:
Q3.8 - Table 1: I suggest presenting the classification data by age group (children and adolescents) and not for each age found.
A3.8 We thank the Reviewer for the suggestion. We have now classified our boys and girls as children and adolescents.
Q3.9 - Explain the differences between the research centers included in the study to clarify the difference in the results found.
A3.9 We thank the Reviewer for the suggestion. Please see A3.7 above. A formal analysis of the between-Center differences (Istituto Auxologico Italiano vs. ICANS vs. Cà Granda) is unfortunately not very meaningful because of the very different samples sizes (see Table 1) and the fact that most cases of severe obesity was provided by the Istituto Auxologico Italiano. Our data are reported, however, in a such way that they can be easily integrated into meta-analyses of CCF, which should disclose the issue of external validation much better than any single study, as we have now pointed out in the conclusion [5].
Q3.10 - Table 1 also contains data for mREE and eREE that were not described in the text. It is necessary to describe the main descriptive findings of mREE and eREE.
Q3.10 We thank the Reviewer for the suggestion. We have now added a general comment about the (eREE-mREE) difference. (Note that the median difference central to our discussion is given in Table 2, as before).
Q3.11 - It is necessary to improve the title of table 1. What are the characteristics of the sample studied that appear in that table? To specify.
A3.11 We thank the Reviewer for the excellent suggestion. The title is now: Table 1. Sex, age, anthropometry and resting energy expenditure of the study population.
Q3.12 - Text about the Table 2: Better describe the interpretation of the data. What are the main results?
A3.10 We thank the Reviewer for the suggestion. We have added a short description of the data at the end of Table 2.
References (for all reviewers)
- Steinberg A.; Manlhiot C.; Cordeiro K.; Chapman K.; Pencharz P. B.; McCrindle B. W.; Hamilton J. K. Determining the accuracy of predictive energy expenditure (PREE) equations in severely obese adolescents. Clin Nutr. 2017;36:1158-1164.
- Müller M. J.; Bosy-Westphal A.; Klaus S.; Kreymann G.; Lührmann P. M.; Neuhäuser-Berthold M.; Noack R.; Pirke K. M.; Platte P.; Selberg O. World Health Organization equations have shortcomings for predicting resting energy expenditure in persons from a modern, affluent population: generation of a new reference standard from a retrospective analysis of a German database of resting energy expenditure. The American journal of clinical nutrition. 2004;80:1379-1390.
- Lazzer S.; Agosti F.; De Col A.; Sartorio A. Development and cross-validation of prediction equations for estimating resting energy expenditure in severely obese Caucasian children and adolescents. Br J Nutr. 2006;96:973-979.
- Bosy-Westphal A.; Braun W.; Schautz B.; Müller M. J. Issues in characterizing resting energy expenditure in obesity and after weight loss. Front Physiol. 2013;4:47.
- Madden A. M.; Mulrooney H. M.; Shah S. Estimation of energy expenditure using prediction equations in overweight and obese adults: a systematic review. J Hum Nutr Diet. 2016;29:458-476.
- Marra M.; Cioffi I.; Sammarco R.; Santarpia L.; Contaldo F.; Scalfi L.; Pasanisi F. Are Raw BIA Variables Useful for Predicting Resting Energy Expenditure in Adults with Obesity. Nutrients. 2019;11
- Bedogni G.; Bertoli S.; Leone A.; De Amicis R.; Lucchetti E.; Agosti F.; Marazzi N.; Battezzati A.; Sartorio A. External validation of equations to estimate resting energy expenditure in 14952 adults with overweight and obesity and 1948 adults with normal weight from Italy. Clin Nutr. 2019;38:457-464.
- Agostoni C.; Edefonti A.; Calderini E.; Fossali E.; Colombo C.; Battezzati A.; Bertoli S.; Milani G.; Bisogno A.; Perrone M.; Bettocchi S.; De Cosmi V.; Mazzocchi A.; Bedogni G. Accuracy of Prediction Formulae for the Assessment of Resting Energy Expenditure in Hospitalized Children. J Pediatr Gastroenterol Nutr. 2016;63:708-712.
- Conroy R. M. What hypotheses do “nonparametric” two-group tests actually test? Stata Journal. 12:182-190.
- Marra M.; Montagnese C.; Sammarco R.; Amato V.; Della Valle E.; Franzese A.; Contaldo F.; Pasanisi F. Accuracy of predictive equations for estimating resting energy expenditure in obese adolescents. J Pediatr. 2015;166:1390-6.e1.
- Shetty P. Energy requirements of adults. Public Health Nutrition. 2005;8:994-1009.
Round 2
Reviewer 1 Report
This is ok now.
This manuscript is a resubmission of an earlier submission. The following is a list of the peer review reports and author responses from that submission.
Round 1
Reviewer 1 Report
This reviewer considers the results of this study as confirmatory. It is not surprising (and has been shown already before) that all REE algorithms addressed have a limitations to predict REE in individuals and populations. This is mainly due to differences between the populations (i.e. the reference population which has been used for the generation of the algorithm and the population studied for validation). Thus, a systematic analysis has to start with a systematic comparison of that two populations. Then technical details and measurement errors have to be taken into account. As a general rule one should not compare algorithms which are based on studies on populations which do not fit to the population of interest. E.g., the Harris Benedict algorithm is based on measurements in adults and no children and adolescents had been included.
Major drawbacks of this study are that neither puberty nor body composition (FFM) and growth rate had been considered as determinants of REE. Since children(i) vary in these characteristics and (ii) FFM is the major determinant of REE one cannot expect a good prediction of REE using age, sex, body weight and height as only predictors.
Both, FM and FFM are determinants of REE, thus, their contribution to REE varies between non obese and obese children. This idea is camouflaged by using BMI as a crude variable.
At the end the authors leave us with the idea of some bias. However the reader is left with the general impression that no algorithm is really good enough to predict REE in non obese and obese children. Would be fine to come out with a solution.
Reviewer 2 Report
Thank you for this paper about the topic REE. I have no comments to the manuscript.
I was just surprised that REE calculated by equations for adults worked so well in children and adolescents. But on the other hand most of your children and adolescents were in the age range of for example Molnar et al..
One important point of REE is the body composition, but there are maybe no data available for your study group.
Reviewer 3 Report
This was a useful paper cross-validating nine equations to estimate REE in a large sample of obese children.
Major:
Please have a table of the 9 equations used (plus reference) and also the population of which they are derived. In particular this will support your statement (line 124-125).
Please do analysis in boys in boys and girls separately as well. Do they produce the same results compared with group analysis in regards to how well the equations predicted measured REE?
Please report baseline RQ values.
2.2. Anthropometry- please explain this more, are these BMI z scores?
Why wasn’t fat and fat free mass collected? Fat free mass is a major predictor of REE so this would also assist with interpretation of data e.g. expressing RMR per kg of FFM.
2.3. REE measurement-
Was physical activity controlled for prior to REE measures?
2.5. Please explain exactly how the CCF was calculated. Was it a fraction converted to a decimal number?
Figure 1- are these percentage CCF? (please make it clear in the legend/axis)
Please rephrase ‘nearly 4 over 10’, what does this mean? Is it meant to be 4 out of 10? (check abstract too).
Some interpretation of this should be included and also future recommendations/research. If we need to be cautious with the Molnar formula, then what do the authors suggest to clinicians in practice to use?
How do these results compare to research done in adults/obese adults? please include some discussion of this in the discussion section.
Minor:
There needs to be a better description of the data in your tables. What do the % in brackets refer to? What do the numbers in brackets refer to? Are they CIs? Median IQR? Abbreviations. Everything needs to be stated in the table footnote. The table presented is hard to read and interpret.
When a table is split across two pages, the column headings need to be inserted again on the second page to make it easier for the reader rather than turn back to the previous page.
Please change wording throughout e.g. instead of obese/overweight children, it should be children with obesity.